# Changes in the Intranetwork and Internetwork Connectivity of the Default Mode Network and Olfactory Network in Patients with COVID-19 and Olfactory Dysfunction

**DOI:** 10.3390/brainsci12040511

**Published:** 2022-04-18

**Authors:** Hui Zhang, Tom Wai-Hin Chung, Fergus Kai-Chuen Wong, Ivan Fan-Ngai Hung, Henry Ka-Fung Mak

**Affiliations:** 1Department of Diagnostic Radiology, The University of Hong Kong, Hong Kong 999077, China; shirlzh7@hku.hk; 2Alzheimer’s Disease Research Network, The University of Hong Kong, Hong Kong 999077, China; 3Department of Microbiology, The University of Hong Kong, Hong Kong 999077, China; tomchungwaihin@gmail.com; 4Department of Ear, Nose and Throat Surgery, Pamela Youde Nethersole Eastern Hospital, Hong Kong 999077, China; wongkaichuen@gmail.com; 5Carol Yu Centre for Infection, The University of Hong Kong, Hong Kong 999077, China; ivanhung@hku.hk; 6The Collaborative Innovation Center for Diagnosis and Treatment of Infectious Diseases, The University of Hong Kong, Hong Kong 999077, China; 7State Key Laboratory of Brain and Cognitive Sciences, The University of Hong Kong, Hong Kong 999077, China

**Keywords:** COVID, olfactory dysfunction, default mode network, olfactory network, functional connectivity, resting-state fMRI

## Abstract

Olfactory dysfunction (OD) is a common symptom in coronavirus disease 2019 (COVID-19) patients. Moreover, many neurological manifestations have been reported in these patients, suggesting central nervous system involvement. The default mode network (DMN) is closely associated with olfactory processing. In this study, we investigated the internetwork and intranetwork connectivity of the DMN and the olfactory network (ON) in 13 healthy controls and 22 patients presenting with COVID-19-related OD using independent component analysis and region of interest functional magnetic resonance imaging (fMRI) analysis. There was a significant correlation between the butanol threshold test (BTT) and the intranetwork connectivity in ON. Meanwhile, the COVID-19 patients with OD showed significantly higher intranetwork connectivity in the DMN, as well as higher internetwork connectivity between ON and DMN. However, no significant difference was found between groups in the intranetwork connectivity within ON. We postulate that higher intranetwork functional connectivities compensate for the deficits in olfactory processing and general well-being in COVID-19 patients. Nevertheless, the compensation process in the ON may not be obvious at this stage. Our results suggest that resting-state fMRI is a potentially valuable tool to evaluate neurosensory dysfunction in COVID-19 patients.

## 1. Introduction

As of 6 March 2022, over 433 million people globally have contracted severe acute respiratory syndrome coronavirus 2 (SARS-CoV-2) [1]. Post-viral olfactory dysfunction (OD) is a critical symptom of coronavirus disease 2019 (COVID-19), with more than 66% of patients in European countries and the US reporting some degree of anosmia [2,3]. In addition to anosmia, many recent studies have shown that many patients with COVID-19 experience central nervous system (CNS) symptoms, including headache, altered mental status, acute cerebrovascular disease, and epilepsy [4,5]. A study published in the Journal of the American Medical Association illustrated that SARS-CoV-2 can negatively affect memory eight months after mild COVID-19 [6]. Therefore, there is an immense need to elucidate the underlying neuronal mechanisms behind these observations.

Using independent component analysis (ICA) and region of interest (ROI)-based methods, multiple spatially distributed large-scale functional brain networks have been detected and named resting-state networks. A number of studies have investigated OD from the perspective of these brain networks [7,8,9]. For example, Kollndorfer et al. showed extended functional connectivity (FC) in the olfactory network (ON) in patients with anosmia after an olfactory training program [8]. Moreover, Chung et al. showed an improvement in mean FC in the ON in patients with COVID-19 who were treated with a combination of oral vitamin A and smell training using electronic portable aromatic rehabilitation diffusers [7]. Another study used resting-state functional MRI (rs-fMRI) to show that FC within default mode network (DMN) regions is sensitive and could serve as a neuroimaging biomarker for neurodegenerative disease [10]. DMN brain regions engage in many high-level cognitive functions, including self-referential processes [11], social cognition [12], episodic memory [13], semantic processing [14,15] and attention [16]. Several studies have also shown that the DMN modulates olfactory processing, suggesting that odour processing can utilise cognitive, attentional and memory resources [17,18]. However, to the best of our knowledge, no studies have evaluated the changes in functional connectivity in DMN with ON in COVID-19 patients with anosmia. Knowledge in this area could advance our understanding of the neuronal mechanisms behind certain COVID-19 manifestations, such as OD.

In this study, we used rs-fMRI, a powerful indicator of spontaneous brain activity, to evaluate the changes in functional intranetwork and internetwork connectivity in the DMN and ON in patients with COVID-19-related OD. We hypothesise that the intranetwork and internetwork connectivity of the DMN and ON differ between patients with COVID-19 and healthy controls (HCs). The results might offer insights into rehabilitative mechanisms and therapy development in COVID-19 patients with OD.

## 2. Materials and Methods

### 2.1. Subjects

A total of 13 healthy adults (age: 45.0 ± 13.2 years old, 7 females/6 males), and 24 COVID-19 patients (age: 43.6 ± 14.0 years old, 14 females/8 males) who presented with persistent (≥3 months) COVID-19-related OD were recruited (Figure 1).

COVID-19 was diagnosed by real-time reverse-transcriptase polymerase chain reaction (RT-PCR) of pooled nasopharyngeal and throat swab specimens targeting the E-gene (TIB Molbiol, Berlin, Germany) of SARS-CoV-2. Patients underwent a complete ear, nose and throat (ENT) examination, and individuals with conductive and sensorineural causes of olfactory dysfunction (e.g., post head injury, cerebrovascular accident, dementia etc.) were excluded. Healthy subjects were recruited from the community by advertisements on campus. Before MR scanning, participants were interviewed to collect their socio-demographic data, history of substance or drug abuse, self-reported alcohol and smoking history, cognitive complaints, past medical history, and related medications. The exclusion criteria included: olfactory dysfunction, head injury, history of stroke, migraine, seizures or cancer within five years. Individuals with active infection, psychiatric diseases, non-ambulatory status, and psychiatric diseases, as well as drug abusers and regular alcohol drinkers [19], were also excluded.

Ethical approval (IRB reference number: UW 20-454, 18 September 2020) was received from the Institutional Review Board of the University of Hong Kong and Hospital Authority Hong Kong West Cluster. Our study was in accordance with the Declaration of Helsinki. Written informed consent was obtained from all participants.

### 2.2. Acquisition of MRI Data

All subjects underwent MRI scans using a 3.0-T scanner (SIGNA Premier; GE Healthcare, Chicago, IL, USA) with a 48-channel head coil.

Structural images were acquired using fast and high-resolution three-dimensional brain volume (BRAVO 3D, T1-weighted inversion-recovery-prepared fast-spoiled gradient recalled echo) postcontrast sequence in the sagittal orientation. The sequence parameters were as follows: repetition time (TR) = 7.3 ms, echo time (TE) = 3 ms, flip angle = 8°, voxel size = 1 × 1 × 1 mm^3^ and field of view (FOV) = 256 × 256 × 170 (mm).

The rs-fMRI data were collected using a gradient-echo echo-planar sequence (T2* weighted) in the axial orientation and were sensitive to blood oxygen level-dependent contrast. The sequence parameters were as follows: TE = 30 ms, TR = 2000 ms, flip angle = 80° and voxel size = 3 × 3 × 4 mm^3^, dummy scans = 0, FOV = 240 × 240 × 152 and 38 slices. During functional scanning, the subjects were instructed to focus on a cross presented in the mirror and to not think of anything. The rs-fMRI data included 180 functional dynamics, which lasted for 6 min.

### 2.3. Quantitative Assessment of Olfactory Function

The butanol threshold test (BTT) and the University of Pennsylvania Smell Identification Test (UPSIT, Sensonics International, Haddon Heights, NJ, USA) were applied as the screening methods for the assessment of olfactory function. Detailed procedures of BTT and UPSIT quantitative measurements and the inclusion criteria have been published previously [20]. Patients were diagnosed with olfactory dysfunction if they met the following criteria: (1) self-reported ongoing olfactory impairment, (2) a BTT score < 4, and (3) an abnormal UPSIT result.

### 2.4. Analysis of rs-fMRI Data

Pre-processing of rs-fMRI data was performed using the Data Processing and Analysis of Brain Imaging (DPABI V6.0 210501; http://rfmri.org/dpabi, accessed on 10 April 2022) toolbox based on Statistical Parametric Mapping software (SPM12, https://www.fil.ion.ucl.ac.uk/spm/software/spm12/, accessed on 10 April 2022). The first ten dynamics were removed, and the remaining dynamics were corrected to the middle slice for the slice-dependent delays. Then, we realigned the images of each participant with the rigid-body linear transformation. If the head motion of a subject was more than 3 mm and 3° in any direction, this subject would be excluded from the following data analysis. Next, the structural images were segmented, and we obtained tissue maps. Using the Diffeomorphic Anatomical Registration Through Exponentiated Lie Algebra tool (DARTEL) [21], the tissue maps and structural images were normalized to the Montreal Neurological Institute (MNI) space. After that, the transformation parameters were generated. After the segmentation, several nuisance signals (i.e., the head motion estimates from the Friston 24-parameter model [22] and the regressors of white matter (WM) and cerebrospinal fluid (CSF)) were regressed out from each voxel’s time series. The images were normalized to the MNI space and resampled to isotropic resolution (3 × 3 × 3 mm^3^) with the transformation parameters derived from DARTEL. Subsequently, all fMRI data were smoothed using a Gaussian kernel (6 mm full width at half maximum) and temporally filtered using the band from 0.01 to 0.1 Hz to reduce the effects of respiratory and cardiac noise. We also removed the linear trend. At last, the scrubbing step was applied. The threshold of framewise displacement (FD) was set at 0.5 mm, and we scrubbed one volume after and two before the motion spike. After confirming that the data had sufficient time series (>4 min), we did the cubic interpolation to prevent temporal leakage of artefact [23,24].

A group ICA was performed for all subjects using the Group ICA of fMRI Toolbox (GIFT v3.0b; https://trendscenter.org/software/, accessed on 10 April 2022). The number of independent components was estimated using the minimum description length criterion. A two-step principal component analysis was used to decompose the dataset into 50 components. The DMN component (component 33) was identified using a spatial mask from a former study [25]. Meanwhile, the component of the ON (component 29) was identified based on former studies [8,26]. Then, a back-reconstruction step was implemented to estimate subject-specific components. The component maps of the ON and DMN for all the participants were separately entered into the one-sample t-test (false discovery rate (FDR)-corrected, *p* < 0.05, voxel size > 2700 mm^3^) to create a sample-specific component map. Furthermore, the seed regions for functional connectivity analyses were defined as the peak value of the respective sample-specific component maps with a 6 mm sphere. The centre of the seed region located at the MNI coordinates as follows: (1) ON: −9, 15, 0 (left caudate nucleus), (2) DMN: −6, −57, 18 (left precuneus). (Figure 2) We calculated the correlation between the ROI series of the two centres (the ON centre and DMN centre) and the whole brain for each subject in a voxel-wise manner. To normalise the distribution of Pearson’s correlation coefficient (r), the values were transformed into standard z scores using the Fisher transformation.

In our results, the ON comprises the frontal gyrus, olfactory cortex, insula, anterior cingulate cortex, hippocampus, amygdala, caudate nucleus, putamen, pallidum and temporal pole. Most of those regions, including the orbital frontal gyrus, olfactory cortex, insula, hippocampus, amygdala, and caudate nucleus, are in line with the literature [26,27].

Our results reported that the regions of the DMN include the superior and middle frontal gyrus, anterior cingulate cortex, middle cingulate cortex, posterior cingulate cortex, hippocampus, parahippocampus, precuneus, cuneus, lingual gyrus, occipital gyrus, fusiform gyrus, angular gyrus, and superior and middle temporal gyrus. Our results are consistent with Andrew-Hanna’s [28] and Li et al.’s studies [29].

Subsequently, the sample-specific component map was applied as the template for its corresponding brain network. Based on Automated Anatomical Labelling (AAL) anatomic parcellation, we obtained 31 ROIs in the ON (coordinates of peak t values in ON regions, sphere radius = 6 mm) (Figure 3A and Appendix A) and 37 ROIs in the DMN (coordinates of peak t values in DMN regions, sphere radius = 6 mm) from the above-mentioned z-score maps (Figure 3B and Appendix A).

The ROI analysis was used to compare network connectivity between the HC group and the COVID-19 group. Each ROI in any given network was independently compared with the other ROIs in the respective network. We generated cross-correlation matrices using Pearson’s correlation (pairwise combination of all 68 ROIs). These individual correlation matrices were subsequently converted into z scores and input into group comparisons within/between the DMN and the ON to investigate differences between HCs and patients with COVID-19.

Statistical analysis was performed with SPSS (version 27, SPSS Inc., Chicago, IL, USA). The sex distribution between the two groups was examined using Pearson’s chi-square test. The two-sample t-test was used to investigate group differences in demographic characteristics and functional connectivity. The relationship between the measurements of smell, as measured by the butanol threshold test (BTT) and the smell identification test (SIT), and internetwork/intranetwork connectivity was calculated using Pearson’s correlation coefficient. A *p*-value of <0.05 was considered statistically significant for all tests.

## 3. Results

A total of 37 subjects were recruited, including 24 patients with COVID-19 and 13 HCs. Two patients’ data were excluded due to head motion during rs-fMRI. The characteristics of COVID-19 patients and HCs are summarised in Table 1. No significant differences in age and sex were observed between the two groups.

Significant differences were observed in the average intranetwork connectivity in the DMN (*p* = 0.013) and in the average internetwork connectivity between the ON and the DMN (*p* = 0.048) (Table 1 and Figure 3) between the two groups. Specifically, patients with COVID-19 had significantly higher functional connectivity than HCs in the DMN (HCs: 0.49 ± 0.10 vs. COVID-19: 0.58 ± 0.10), as well as higher internetwork connectivity between the ON and the DMN (HCs: 0.21 ± 0.09 vs. COVID-19: 0.28 ± 0.09). There is no statistical difference in the average intranetwork connectivity in the ON (*p* = 0.20) between patients and HCs (Table 1 and Figure 4).

In addition, the correlation between the BTT score and the average intranetwork connectivity in the ON demonstrated a significantly positive correlation (r = 0.499 *, *p* = 0.025), which could indicate a relationship between clinical olfactory performance and intranetwork connectivity in ON (Figure 5A). No significant correlation was found between the SIT score and the average intranetwork connectivity in ON (r = 0.367, *p* = 0.112).

## 4. Discussion

Our findings demonstrate greater activity at rest within the DMN and an enhanced association between the DMN and the ON in patients with COVID-19. There was no significant difference in activity within the ON between HCs and COVID-19 patients. In addition, a positive statistical correlation was observed between the BTT score and the average intranetwork connectivity in the ON.

In previous studies, the BTT and UPSIT scores in HCs were significantly higher than in patients with COVID-19 [20,30]. However, the precise pathogenesis of OD in patients with COVID-19 remains uncertain. Some histological studies have shown that SARS-CoV-2 infection is localised at the olfactory neuroepithelium, which may disrupt biochemical and electrophysiological homeostasis [20,31]. Furthermore, evidence shows that SARS-CoV-2 affects the olfactory system and the CNS [32]. SARS-CoV-2 can enter the CNS by crossing the neural–mucosal interface in the olfactory mucosa and penetrating neuro-anatomical structures associated with the primary respiratory and cardiovascular control centres [33]. Poyiadji et al. reported a case of a patient with acute necrotising encephalopathy associated with COVID-19 [34]. Another study detected the SARS-CoV-2 genome in CSF [35]. In our study, the recruited patients with COVID-19 had experienced OD for more than three months. We infer that the changes in functional connectivity within/between the DMN and the ON were sequelae of SARS-CoV-2 infection.

Some studies have investigated the role of the DMN in olfactory processing. In a task-based fMRI study, direct connectivity between the DMN and the ON was identified in an odour–visual association paradigm, suggesting that olfactory perception may utilise cognitive, memory and attentional resources during odour tasks [17]. Another study discovered reduced connectivity between the ON and the DMN in Alzheimer’s disease (AD) during an odour–visual association task. That study provided evidence for the selective vulnerability of the ON and the DMN in patients with AD. It also supported the important role of the DMN in olfactory perception [18]. Carlson et al. reported results that illustrated the prolonged impact of odours with different affective valences on mood and cognitive function using rs-fMRI [36]. Interestingly, a recent study (N = 496) examined negative affect (NA) during the COVID-19 outbreak and after the peak of the pandemic. NA did not decrease in participants even after the peak; however, individuals with higher connectivity within the salience network (SN) and between the SN and the DMN or frontoparietal network showed less NA during and after the peak of the pandemic compared with before the pandemic [37]. Our results showed a significantly higher value in the average internetwork connectivity of the COVID group between ON and DMN, which provided evidence for the stronger engagement of DMN by the ON in COVID patients. However, we did not find any significant correlation between olfactory scores and the intranetwork connectivity of DMN/the internetwork connectivity between DMN and ON. Our study has a relatively small size in rs-fMRI research. Thus, to further prove the efforts of the DMN in the regulation of olfaction in COVID patients, a larger sample scale shall be considered.

We hypothesise that the observed increase in connectivity at rest may be compensatory; that is, it may be an attempt to compensate for deficits in olfactory processing and general well-being in patients with COVID-19. A large number of studies have investigated the critical role of the DMN in healthy ageing adults [10,38], apolipoprotein E4 carriers [39,40] and patients with AD [41]. Numerous theories were successively put forward to explain the altered brain activities in cognitive aging issues [38]. Increases in DMN connectivity have been reported in mild cognitive impairment and AD patients compared with HCs [10]. Taken together, these results indicate the existence of compensatory processes in the early stage of AD. Unambiguously, the detection of the compensation in preclinical neurodegenerative diseases would help delay the onset of clinical diseases and sustain the overall functioning of the brain. Similar to the above-mentioned studies, our study shows the intranetwork connectivity in the DMN of COVID-19 patients, which indicates the compensation for OD has a significantly higher increase than HCs and a statistical increase of the average internetwork connectivity between DMN and ON in patients compared with HCs.

Nonetheless, no significant difference was found in the intranetwork connectivity in the ON. A newly published longitudinal study of SARS-CoV-2 uses a very large sample size (HCs: N = 384; COVID-19 patients: N = 401). Days of infection before the second-time scan is from 35 to 407 days. In the longitudinal group comparison, a significant increase in diffusion measures for the COVID group was found in the imaging-derived phenotypes within olfactory subnetworks. The patients in the second time point showed a cognitive decline compared with the first time [42]. Another study investigated the changes of four small-world graphical metrics between SARS-CoV-2-infected subjects (N = 27, hyposmia, range 11–89 days) and healthy controls (N = 18). Both structural and functional connectivity metrics in patients presented significantly higher values than HCs [43]. Clearly, different from their studies, our cohort recruited long COVID patients with OD. The time from OD onset to MR scans of the participants ranged from 91 to 268 days. The small sample size might be another reason that affects the results, as the average intranetwork connectivity in the ON has a high amount of variability (larger standard deviation) in the healthy controls. Referring to Postma et al.’s study [44], we have calculated the correlation between the duration of the olfactory disfunction and the averaged functional network connectivity in the ON, in the DMN and between these two networks. However, no significant correlation (*p* > 0.05) was seen. Douaud et al.’s study had a larger sample size and follow-up (duration of infection: from one to thirteen months), but they also did not discover any significant effect.

Moreover, there is a positive statistical correlation between BTT and the intranetwork connectivity in the ON. BTT is a test where the odour detection threshold is measured, whereas UPSIT is the most commonly used test for odour identification [45]. This finding demonstrates a close relationship between the extent of odour detection and intranetwork connectivity in the ON, which suggests that intranetwork connectivity in the ON might be useful to evaluate olfaction in patients with COVID-19.

This study has some limitations that should be noted. First, on account of the limited manpower, the quantitative olfactory assessments were only performed in patients with COVID-19. To lessen the impact, we recruited HCs who self-reported that they were without OD symptoms. Second, we did not collect any neurological data other than olfactory disfunction. Previous studies have demonstrated that cognition (i.e., attention [17] and memory [18]) and emotions (i.e., pleasantness scores [36] and eleven negative emotions experienced based on the NA questionnaire [37]) have close relationships with the functional connectivity within and between the ON and DMN. It would therefore be worthwhile to include those assessments in the further study. Third, since our study had a relatively small sample size, further large-sample studies will be needed to clarify our observations in the future.

## 5. Conclusions

In conclusion, our results suggest that significantly higher intranetwork connectivity in the DMN and an enhanced trend between the DMN and the ON compensate for the deficit in olfactory processing and recovery of general well-being in patients with COVID-19. Moreover, the insignificant increases in patients regarding the intranetwork connectivity within the ON may indicate that the compensation process is not obvious at this stage. Some specific treatments would be necessary for recovery. Still, these should be validated in a larger-scale study.

## Figures and Tables

**Figure 1 brainsci-12-00511-f001:**
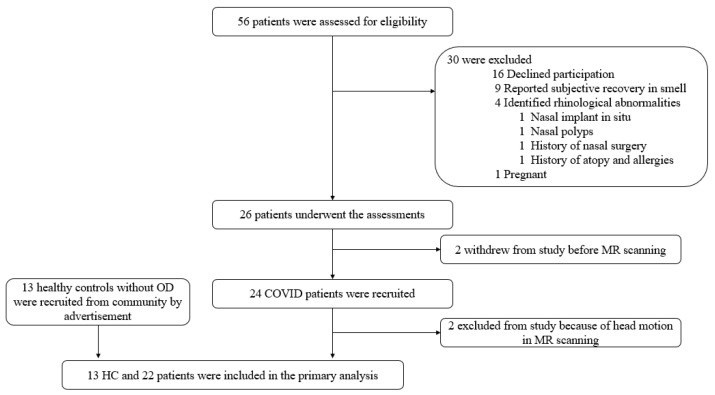
Flow diagram showing the subject recruitment process in this study.

**Figure 2 brainsci-12-00511-f002:**
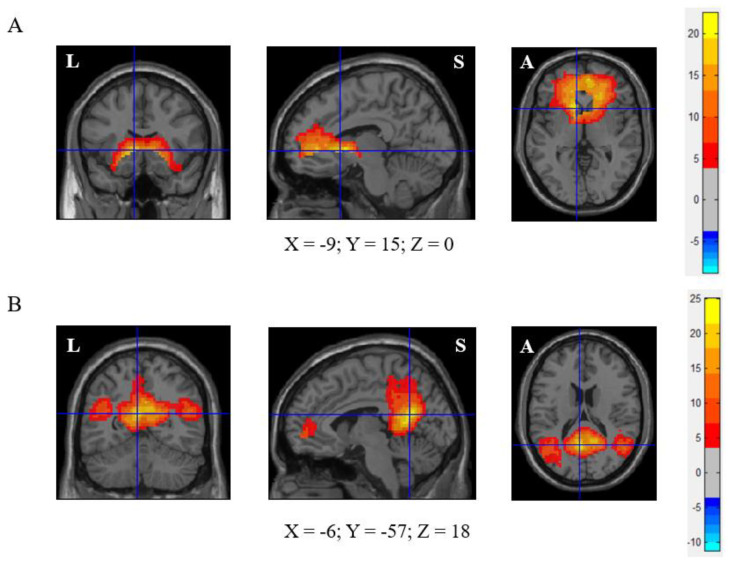
Olfactory network (ON) and default mode network (DMN) of this cohort. (**A**) ON (left caudate nuclei (−9, 15, 0) as seed), (**B**) DMN (left precuneus (−6, −57, 18) as seed). FDR correction, *p* < 0.05, voxel size > 2700 mm^3^.

**Figure 3 brainsci-12-00511-f003:**
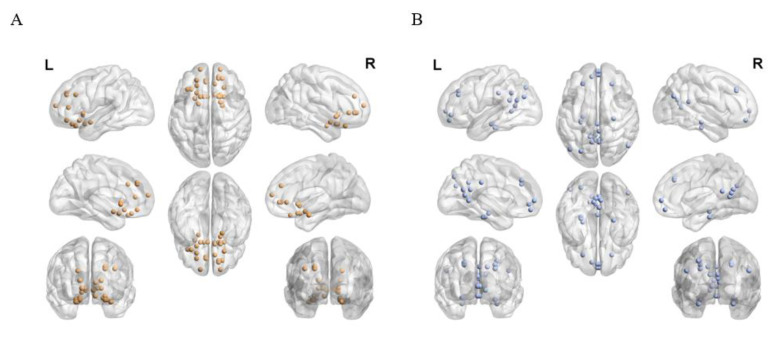
(**A**) A total of 31 ROIs from ON and (**B**) 37 ROIs from DMN are used for ROI-wise analysis between/within ON and DMN.

**Figure 4 brainsci-12-00511-f004:**
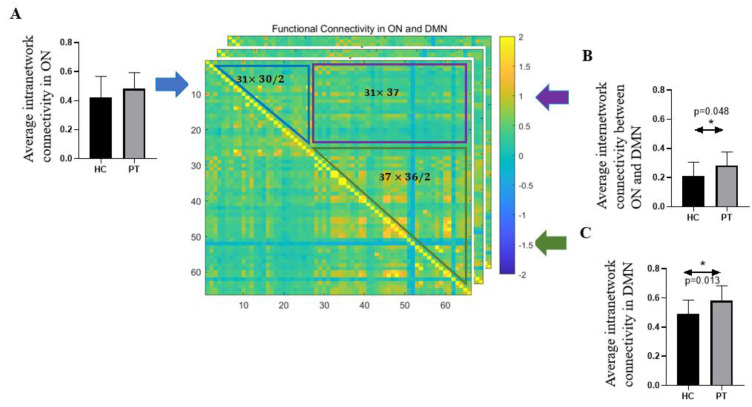
Schematic representation of the comparison of average intra- and inter-network connectivity in ON and DMN between COVID patients and healthy adults. Each subject has a 68 × 68 functional connectivity matrix. The values in the blue triangle indicate the correlation coefficients between pairs of the 31 regions within the ON, and in the green triangle represent the functional connectivity between the corresponding regions in DMN (37 regions). Voxels in the purple rectangle illustrate the functional connections between regions in ON and regions in DMN. In healthy control and patients, the comparison of (**A**) average intranetwork connectivity in ON; (**B**) average internetwork connectivity between ON and DMN and (**C**) average intranetwork connectivity in DMN. (Error bar shows the standard deviation of measurements, * indicates that *p* < 0.05).

**Figure 5 brainsci-12-00511-f005:**
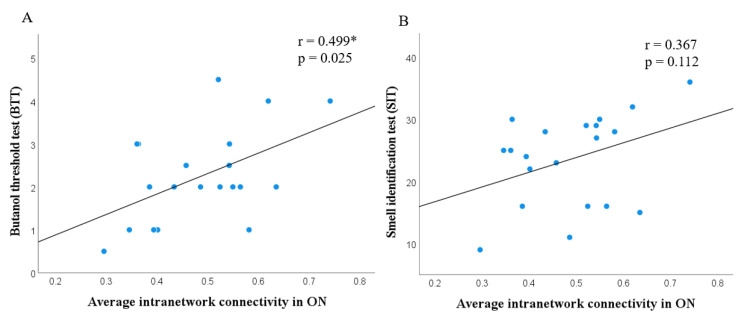
Scatter plot showing correlation between the average intranetwork connectivity in olfactory network and butanol threshold test (BTT) (*r* = 0.499 *, *p* = 0.025); smell identification test (SIT) (*r* = 0.367, *p* = 0.112) in COVID-19 patients.

**Table 1 brainsci-12-00511-t001:** Demographic information of this study.

Characteristics	Healthy Controls	COVID-19 Patients	*p* Value
N	13	22	-
Butanol threshold test (BTT)	-	2.25 ± 1.09	-
The University of Pennsylvania Smell identification test (UPSIT)	-	23.6 ± 7.4	-
OD-onset to MR scan (Days)	-	164.2 ± 50.6	-
SARS-CoV-2 diagnosis	negative	positive	-
Average intranetwork connectivity in ON	0.42 ± 0.14	0.48 ± 0.11	0.20
Average intranetwork connectivity in DMN	0.49 ± 0.10	0.58 ± 0.10	0.013 *
Average internetwork connectivity between ON and DMN	0.21 ± 0.09	0.28 ± 0.09	0.048 *

* indicates *p* < 0.05.

## Data Availability

The data presented in this study are available on request from the corresponding author.

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
