# Peer review of "Changes in the Intranetwork and Internetwork Connectivity of the Default Mode Network and Olfactory Network in Patients with COVID-19 and Olfactory Dysfunction"

_brainsci, 2022, doi:10.3390/brainsci12040511_

Round 1

Reviewer 1 Report

Overall comment

The paper addresses possible changes in the brain connectivity of patients affected olfactory disfunction due to COVID-19. The manuscript is well written and the methodology is clearly stated. However, some points in the Methods should be more carefully addressed to improve the clarity of the paper and its link with the previous literature. Similarly, additional relatively simple analyses could strengthen the results and help their interpretation.

  • Demographic data of the participants, such as sex and age, should be removed from Table 1 and reported separately in the paragraph 2.1 to improve the clarity of the paper.
  • It is not completely clear if all COVID-19 patients tested negative at the RT-PCR test at the scanning day
  • As resting state fMRI data were used, it would be appropriate to check and account for also the motion associated with the spike in the time series by assessing the instantaneous framewise displacement (Power et al., 2014; Satterthwaite, Elliott, et al., 2013).
  • While it is relatively easy to individuate the DMN among the output of an ICA analysis, the selection of other less common network, such as the ON, is not that straightforward. The authors stated that ON was extracted from the ICA components by an experienced researcher (p. 170, l. 159-160). However, to improve the clarity of the paper the authors should report at least a reference and the criteria on which they have based their selection of the ON.
  • A list of the anatomical regions and their centre of coordinates included in the two networks would increase the reproducibility of the paper.
  • Could the author clarify how they created a sample-specific component map (p. 170, l.160-162) and why they used a false discovery rate at p<0.01 instead of the more common threshold of p=0.05?
  • The authors should report if the ROIs of both networks are in line with the previous extant literature (see e.g., the recent work https://doi.org/10.1523/ENEURO.0551-19.2020 for the ON).
  • In paragraph 2.4 it has been written that “The relationship between the measurements of smell, as measured by the butanol threshold test (BTT) and the smell identification test (SIT), and internetwork/intranetwork connectivity was calculated using Pearson’s correlation coefficient”. However, with the exception of the BTT score, no results are reported for the SIT. Even if the results are not significant, the authors should report them.
  • Please, when addressing studies about COVID-19 brain related changes consider to cite other relevant recent works such as, e.g., https://www.nature.com/articles/s41586-022-04569-5 and https://onlinelibrary.wiley.com/doi/full/10.1002/hbm.25741 that specifically investigated the olfactory network.
  • I think that the interpretation of the findings is a bit too speculative. The authors do not have data to affirm that “that stronger engagement of the DMN by the ON in our study might indicate the efforts of the DMN to regulate olfactory processing to resist NA.” Moreover, the higher correlation between ON and DMN is not statistically significant, as there is only a “trend”.
  • As the authors interpreted the increase of the DMN intranetwork connectivity as a way to compensate the deficits in the olfactory processing. I think that their interpretation could be boosted by the investigation of the correlation of the DMN intranetwork with the olfactory scores
  • 9 l. 282-284: I think that the first sentence is incomplete. Maybe the period between the two sentences should be changed with a comma.
  • 9 l. 290-294: I do not get the link between the significant correlation between ON connectivity and the BTT score and the inter-network connectivity between ON and DMN. As the authors did not test the correlation of this internetwork connectivity with the BTT, they do not have the data to claim this link.
  • If I have understood correctly, the authors have the data about the duration of the olfactory dysfunction due to COVID-19. It would be interesting to correlate these data with the connectivity in the ON, in the DMN and between the two and it could help the interpretation of the role of the DMN in compensating the olfactory disfunction. For instance, a recent study https://www.nature.com/articles/s41598-021-92224-w observed a correlation with the duration of the olfactory disfunction and brain morphological changes.

Reviewer 2 Report

Abstract:

  1. The number of participants is low for the rs-fMRI study.
  2. I would like to suggest using the word ´potentially valuable tool´ in line 30. 

Results

  1. Mean- while, the average internetwork connectivity between the DMN and ON demonstrated an increasing trend in patients compared with HCs (p = 0.07).
    1. Please detail what does the author mean by increasing trend?
    2. Please also clarify which regions are affected in detail.
    3. This is too general.
  2. Please define regions involved for intranet works and internetworks
  3. Figure no 4, please explain in detail what it is all about.
  4. The authors did mention the neurological deficit in the introduction, but none of these data was collected. Authors should include this neurological data if there is any.

Discussion

  1. Our findings demonstrate greater activity at rest within the DMN and an enhanced interplay trend between the DMN and the ON in patients with COVID-19
    1. Very difficult to understand this sentence. Please revert.
  2. The overall discussion needs to be rewritten, focusing on the main findings and discuss the main finding.
  3. There are discrepancies between the current study results and the previous study. Authors should discuss the number of the sample size in the previous studies, this could be the reason for the differences.
  4. Please include the association or disassociation between UPSIT and rs-fMRI result. 

Conclusion

  1. Do you think it is too early to suggest for compensation mechanism? Need to consider too – number of the sample size is too small for rs-fMRI study.

Round 2

Reviewer 1 Report

The authors did a good work to address all my comments. I have no other questions.

There is a typo in the description of the framewise displacement procedure. In the paper the threshold is "0.05" instead of 0.5 (as recommended in literature and written in the authors' response)